# Radiosensitization by Hyperthermia: The Effects of Temperature, Sequence, and Time Interval in Cervical Cell Lines

**DOI:** 10.3390/cancers12030582

**Published:** 2020-03-03

**Authors:** Xionge Mei, Rosemarie ten Cate, Caspar M. van Leeuwen, Hans M. Rodermond, Lidewij de Leeuw, Dionysia Dimitrakopoulou, Lukas J. A. Stalpers, Johannes Crezee, H. Petra Kok, Nicolaas A. P. Franken, Arlene L. Oei

**Affiliations:** 1Laboratory for Experimental Oncology and Radiobiology (LEXOR), Center for Experimental Molecular Medicine, Amsterdam University Medical Centers, P.O. Box 22700, 1100 DE Amsterdam, The Netherlands; m.xionge@amsterdamumc.nl (X.M.); r_ten_cate@hotmail.com (R.t.C.); h.rodermond@amsterdamumc.nl (H.M.R.); lidewij.deleeuw@gmail.com (L.d.L.); d.dimitrak@hotmail.com (D.D.); l.stalpers@amsterdamumc.nl (L.J.A.S.); n.a.franken@amsterdamumc.nl (N.A.P.F.); 2Department of Radiotherapy, Amsterdam University Medical Centers, P.O. Box 22700, 1100 DE Amsterdam, The Netherlands; casparvanl@hotmail.com (C.M.v.L.); h.crezee@amsterdamumc.nl (J.C.); h.p.kok@amsterdamumc.nl (H.P.K.)

**Keywords:** human papillomavirus, ionizing radiation, hyperthermia, time interval, sequence

## Abstract

Cervical cancers are almost exclusively caused by an infection with the human papillomavirus (HPV). When patients suffering from cervical cancer have contraindications for chemoradiotherapy, radiotherapy combined with hyperthermia is a good treatment option. Radiation-induced DNA breaks can be repaired by nonhomologous end-joining (NHEJ) or homologous recombination (HR). Hyperthermia can temporarily inactivate homologous recombination. Therefore, combining radiotherapy with hyperthermia can result in the persistence of more fatal radiation-induced DNA breaks. However, there is no consensus on the optimal sequence of radiotherapy and hyperthermia and the optimal time interval between these modalities. Moreover, the temperature of hyperthermia and HPV-type may also be important in radiosensitization by hyperthermia. In this study we thoroughly investigated the impact of different temperatures (37–42 °C), and the sequence of and time interval (0 up to 4 h) between ionizing radiation and hyperthermia on HPV16^+^: SiHa, Caski; HPV18^+^: HeLa, C4I; and HPV^−^: C33A, HT3 cervical cancer cell lines. Our results demonstrate that a short time interval between treatments caused more unrepaired DNA damages and more cell kill, especially at higher temperatures. Although hyperthermia before ionizing radiation may result in slightly more DNA damage, the sequence between hyperthermia and ionizing radiation yielded similar effects on cell survival.

## 1. Introduction

Most cases of cervical cancer are caused by infection with human papillomavirus (HPV). HPV types 16 and 18 are responsible for >70% of all cases of cervical cancer in the world [1]. HPV infections are the major risk factor of this disease, which, worldwide, is the second most common form of cancer in women. The combination of radiotherapy and cisplatin-based chemotherapy is presently a standard treatment for patients suffering from cervical cancer. However, if patients have contraindications for cisplatin-based chemotherapy, radiotherapy can be combined with hyperthermia. Mild hyperthermia (i.e., heating the tumor to 40–42.5 °C for 1 h) has been used in the clinic to sensitize radiotherapy since the 1980s, with excellent results for various tumor types [2,3], including cervical cancer [4,5,6]. Radiotherapy plus hyperthermia is a good alternative for women with contraindications for chemotherapy, usually because of limited renal function or frailty, as the survival after daily radiotherapy plus weekly hyperthermia is similar to that of chemoradiotherapy in cervical cancer [7,8].

Hyperthermia has the ability to change the microenvironment and make cells more sensitive to radiotherapy. The precise mechanisms on how hyperthermia enhances tumor oxygenation are not fully understood. Hyperthermia can induce hypoxia-inducible factor-1 (HIF-1), a potent regulator of tumor vascularization and metabolism, which can push surviving cells toward glycolysis [9]. Hyperthermia can indeed increase glycolysis and subsequently reduce oxygen consumption rate [10,11,12]. Thus hyperthermia has been found to enhance tumor perfusion and decrease oxygen consumption, and as a result, hyperthermia reduces tumor hypoxia [9]. Since simultaneous hyperthermia and radiotherapy is technically challenging, common clinical practice is to deliver hyperthermia and radiotherapy sequentially. A major advantage of mild hyperthermia is that it induces no side effects and hardly increases radiation-induced side effects if given sequentially [6]. However, there is no consensus on the optimal sequence and time interval in applying radiotherapy and hyperthermia. In the clinic, hyperthermia is usually given shortly before or after radiotherapy. Moreover, hyperthermia is only applied in a limited number of clinical centers. Therefore, patients who receive radiotherapy in a different center than where their hyperthermia treatment is applied may have several hours of travel time between receiving radiotherapy and hyperthermia. This can result in a longer time interval between radiotherapy and hyperthermia than when both treatments are applied in the same center. An important question is whether it remains beneficial to give the patient the daily fractionated radiotherapy in a facility nearby and have her travel once a week to a more distant hyperthermia facility, usually several hours from home, or that both modalities should be given with the shortest interval possible in one dedicated hyperthermia and radiotherapy center. 

Multiple mechanisms are responsible for the radiosensitization effect by hyperthermia, each operating at different temperatures and each with different optimal sequences and time intervals [13,14]. Hyperthermia combined with radiotherapy improves treatment outcome, which can be explained by multiple factors. First, preheating the tumor induces reoxygenation by enhancing blood flow and by reducing oxygen consumption [15,16,17]. Second, hyperthermia interferes with the homologous recombination, preventing the radiation-induced DNA breaks from being repaired [18,19]. Homologous recombination requires a sister chromatid and is thus mainly active during the S-phase and G2-phase of the cell cycle and is considered error-free [20]. This repair pathway is therefore also indicated as the slow repair pathway as it usually repairs the more complicated DNA double strand breaks (DSBs) [21]. Due to the fact that radiotherapy and hyperthermia are both local therapies, an accumulation of DNA damage will result in selective eradication of more cancer cells. Simultaneous application of radiotherapy and hyperthermia will cause the highest amount of DNA damage, not only in the tumor but also in the normal tissue of the locally treated area [21]. Sequential application of radiotherapy and hyperthermia is considered tumor-selective, as this is thought to cause significantly less cell kill in normal cells compared to tumor cells [22]. Third, hyperthermia can radiosensitize cancer cells in acidic, hypoxic, and nutrient-deprived tumor areas [23,24,25]. Sequence and time interval are not relevant for this mechanism, but the temperature level is. Fourth, hyperthermia has a specific antitumor effect in HPV-positive cervical cancer cells [26]. 

In this study we investigated in in vitro models of HPV16^+^ (SiHa and Caski), HPV18^+^ (HeLa and C4I), and HPV negative (C33A and HT3) cell lines, which sequence and time interval (0 up to 4 h) between applying ionizing radiation and hyperthermia caused the most damage to the tumor cells at different temperatures (37 to 42 °C). Effects on cell cycle distribution, DNA damage level, and cell survival fraction were assessed. Finally, different mechanisms explaining these results will be discussed.

## 2. Results

### 2.1. A Short Time Interval between Ionizing Radiation and Hyperthermia Results in a Lower Cell Survival

To evaluate the effects of the sequence and the time interval between ionizing radiation and hyperthermia on cell survival, clonogenic assays were performed after hyperthermia or ionizing radiation only or hyperthermia followed by ionizing radiation and after the reversed order. Furthermore, different time intervals between ionizing radiation and hyperthermia (0, 2, and 4 h) were tested and the effect of different temperatures was investigated for HPV16^+^, HPV18^+^, and HPV-negative cervical carcinoma cell lines. A schematic overview of treatments is given in Figure 1A. The heat maps presented in Figure 1B–D demonstrate the survival fraction after each treatment, a darker color is related to lower cell survival for the following cell lines: HPV16^+^ SiHa (red) and Caski (orange) (Figure 1B); HPV18^+^ HeLa (green) and C4I (yellow) (Figure 1C); and HPV-negative C33A (blue) and HT3 (turquois) (Figure 1D). Per the heat map, from top to bottom, the dose of ionizing radiation increased (2, 4, 6, and 8 Gy) and per dose of ionizing radiation elevated levels of hyperthermia were applied (37, 39, 41 and 42 °C). Per the heat map, from left to right, the time between ionizing radiation and hyperthermia is shown, e.g., on the left side first hyperthermia before ionizing radiation with 4, 2, or 0 h time interval, and on the right side of the heat map hyperthermia was applied after ionizing radiation, with the same time intervals between the treatments. First, the heat maps demonstrate that a higher dose of ionizing radiation is correlated with lower cell survival. Second, cells treated at a higher temperature showed a lower survival fraction. Third, from the outside towards the middle columns, the color gets darker, indicating a lower survival fraction after treatments with a shorter time interval between two therapies. Fourth, HPV-negative cell lines showed lower survival fractions than the HPV-positive cell lines at lower doses of ionizing radiation. Lastly, combined hyperthermia with ionizing radiation resulted in a lower cell survival fraction compared to ionizing radiation only. 

Cells treated with ionizing radiation alone are presented in Figure 1E–G, demonstrating the survival fraction after ionizing radiation only and after hyperthermia combined with ionizing radiation with a 0 hour and 4 hour time interval for both sequences. HPV16^+^ SiHa (red) and Caski (orange) (Figure 1E); HPV18^+^ HeLa (green) and C4I (yellow) (Figure 1F); and HPV-negative C33A (blue) and HT3 (turquois) (Figure 1G) cervical cancer cell lines. We demonstrated that for all cell lines the survival fractions were lower when hyperthermia was added to ionizing radiation. Moreover, in these graphs a shorter time interval (0 h) resulted in a lower survival fraction than a long time interval (4 h) between the two modalities. The order of applying ionizing radiation and hyperthermia did not have any differences in terms of survival fraction. 

The heat maps of all six cervical cancer cell lines, as shown in Figure 1B–D, illustrate that a higher temperature and a higher dose of ionizing radiation resulted in a lower survival fraction compared to a lower temperature or a lower dose of ionizing radiation. In Appendix A, bar graphs of at least four independent experiments show means with standard deviation of clonogenic cell survival of HPV16^+^ cell lines (Appendix A) SiHa (red), Caski (orange); (Appendix A) HPV18^+^ cell lines HeLa (green), C4I (yellow), HPV-negative cell lines (Appendix A) C33A (blue), and HT3 (turquoise). For each cell line, the top lane shows the results of hyperthermia prior to ionizing radiation, the bottom lane shows the results of hyperthermia after ionizing radiation, and for each sequence also, from left to right, the results after different hyperthermia temperatures (39, 41, and 42 °C) and different ionizing radiation doses (0, 2, 4, 6, and 8 Gy). As confirmed in Appendix A, the statistical analysis depicted in asterisks above the bar graphs shows that in most cases a longer time interval between the two therapies resulted in a higher survival fraction compared to a shorter time interval. Moreover, combined hyperthermia with ionizing radiation correlated with a lower cell survival fraction compared with ionizing radiation only. 

In conclusion, our results showed that the shortest time interval (0 h) between ionizing radiation and hyperthermia resulted in the lowest cell survival. Moreover, there is no significant difference between the different sequences in clonogenic capacity.

### 2.2. A Higher G2 Arrest after a Short Time Interval between Hyperthermia and Ionizing Radiation

Cell cycle distribution was studied by the incorporation of BrdU. A schematic overview of treatments is given in Figure 2A. Means with standard deviation of at least four replicates are presented (Figure 2B–D and Appendix A). In all cell lines, different time intervals between ionizing radiation and hyperthermia (0, 1, 2, 3, and 4 h) were tested, the temperature of hyperthermia (42 °C) and ionizing radiation dose (4 Gy) were investigated for HPV16+ (Figure 2B) HPV18+ (Figure 2C), and HPV-negative (Figure 2D) cell lines. Untreated samples had approximately 40–55% of cells in G1 phase, around 20–35% in S phase, and only 5–10% in G2 phase. Cells treated with a short time interval had the highest increase in the proportion of cells in G2 phase compared to the untreated samples (ctrl), and a decrease of cells in G1 phase was observed after treatments with ionizing radiation and hyperthermia. There were no significant differences observed between hyperthermia prior to ionizing radiation or hyperthermia after ionizing radiation. Cell cycle distribution was more dependent on the cell line than on the type of HPV.

### 2.3. Apoptotic Levels are the Highest after a Short Time Interval between Ionizing Radiation and Hyperthermia 

Apoptosis after different time intervals and for the different sequences of ionizing radiation (4 Gy) and hyperthermia (42 °C) was measured by the Nicoletti assay. A short time interval between ionizing radiation and hyperthermia resulted in the highest apoptotic levels measured by flow cytometry. All cell lines showed that approximately 30–35% of cells were apoptotic after a 0 h time interval between ionizing radiation and hyperthermia, compared to approximately 20–30% after a 4 h time interval between ionizing radiation and hyperthermia (Figure 3). All cell lines showed that a shorter time interval between the two treatment modalities induced more apoptosis. Compared with ionizing radiation only, which is the dotted line in Figure 3D–F, combined hyperthermia with ionizing radiation induced a higher level of cell apoptosis. When cells were treated with ionizing radiation alone, the apoptosis levels were cell line dependent and were in the range of 5–15%. When the apoptosis levels of ionizing radiation alone were below 10%, a clear trend was observed that after the combination of hyperthermia and ionizing radiation, a short time interval resulted in a higher apoptosis level. However, when the apoptosis levels after ionizing radiation alone were above 10%, there were no significant differences found between a short or a long time interval between hyperthermia and ionizing radiation. Moreover, the sequence of applying ionizing radiation and hyperthermia did not result in any significant differences in apoptosis levels.

### 2.4. γ-H2AX Foci Levels Are Increased at Higher Temperatures and at Shorter Time Intervals between Ionizing Radiation and Hyperthermia

DNA damage, specifically DNA double strand breaks (DSBs), was measured by nuclear γ-H2AX staining, which may be very important to understand the earlier described differences in cell survival. To prevent the counting of nonradiation induced DSBs, exclusion of S-phase cells was done using Edu-exclusion (Figure 4A). In Figure 4B, representative pictures of SiHa cells demonstrate a higher number of γ-H2AX foci after a 1 hour treatment with 42 °C than after 39 and 41 °C combined with 2 Gy of ionizing radiation. A shorter time interval between the two therapies resulted in more DSBs than after a 4 hour time interval, especially after a 42 °C heating combined with ionizing radiation. There was no significant difference when hyperthermia preceded or followed ionizing radiation. The dot plots graphs demonstrate the number of foci counted per nucleus for HPV16^+^ SiHa and Caski (Figure 4C), HPV18^+^ HeLa and C4I (Figure 4D), and HPV-negative C33A and HT3 (Figure 4E) cell lines. Per cell line, we treated cells with three different temperatures, four different time intervals, and both ionizing radiation prior to hyperthermia or the reversed sequence. The trend is pretty clear among all cell lines, for all temperatures: a short time interval resulted in more residual DNA breaks at 24 h after treatment. Levels of γ-H2AX foci are higher after hyperthermia at 42 °C compared to 39 and 41 °C. 

The HPV16^+^ cell lines (Figure 4C) demonstrate that after heating at 39 °C, either ionizing radiation immediately before or after hyperthermia results in the highest number of γ-H2AX foci, while after heating at 42 °C, a 0 h time interval between ionizing radiation and hyperthermia was significantly better than after a longer time interval. Remarkably, only for the HPV16^+^ cell lines, we observed that when applying hyperthermia prior to ionizing radiation, at least for heating at 42 °C, no significant differences were found in the number of γ-H2AX foci between the different time intervals. For HPV18^+^ cell lines (Figure 4D), a short time interval between ionizing radiation and hyperthermia also resulted in higher γ-H2AX foci levels compared to a long time interval. Differences were more pronounced after heating at higher temperatures. In HPV negative cell lines (Figure 4E) the time interval between ionizing radiation and hyperthermia significantly induced more DNA breaks for any sequence and after any temperature of heating on both cell lines. All cells were fixed and stained at 24 h after treatment, indicating that there is more remaining DNA damage after a short time interval between the two therapies. Overall, a higher hyperthermia temperature resulted in higher γ-H2AX foci levels.

## 3. Discussion

The results of the present in vitro study demonstrated that the sequence of ionizing radiation and hyperthermia makes no major difference in terms of radiosensitization, since both sequences induced similar amounts of cell survival, similar changes in cell cycle distributions, similar levels of remaining DNA breaks, and similar changes in apoptotic levels (Figure 5). In our previous publication, we have already demonstrated the additional effect of adding hyperthermia to ionizing radiation [26]. In this study, we focused on the effect of time intervals between, and sequences of hyperthermia and ionizing radiation on, DNA damage and cell survival. However, according to a retrospective study on cervical cancer patients, it is of clinical relevance to keep the time interval as short as possible between hyperthermia and ionizing radiation to increase tumor cell kill, which may be pivotal in tumor control and patient cure [27]. This significant effect of shorter intervals can be explained by an increase of cells with unrepaired DNA damage, resulting in more apoptosis.

Cell cycle check-points prevent cells from being replicated or divided if DNA is damaged, by prolonging cell cycle phases. There were only slight changes in the cell cycle distribution, which can indicate the presence of DNA damage, but this was cell line dependent. In all six cervical cancer cell lines, a short time interval led to a higher percentage of cells in the G2 phase, indicating that cells cannot pass the last checkpoint before cell division due to compromised DNA integrity. The percentage of cells in S phase increased, in some cells, after longer time intervals (4 h) between the two treatment modalities, suggesting cells are halted and the damage is too severe to continue to the G2 phase. Since the number of cells in S phase was unchanged or larger after treatment, there is no indication that cells went into cell senescence. The outcome of ionizing radiation is affected by the cell cycle, mitotic cells are hypersensitive because they can inactivate DSB repair, so during interphase, cell survival is maximal when cells are irradiated during the early G1 and G2 phase of the cell cycle and is minimal during the S phase [28]. Previous studies have claimed that this trend is cell line dependent [29]. 

Higher levels of DNA damage were observed after treatment with a shorter time interval between ionizing radiation and hyperthermia, indicating that there was more remaining DNA damage after a short time interval between the two therapies. Blocking DNA repair was clearly the dominant radiosensitizing effect of hyperthermia in our in vitro experiments, the question is: which pathways are involved? Currently, it is generally accepted that hyperthermia downregulates BRCA2 [30], a protein of the homologous recombination (HR). As the HR requires a sister chromatid [20], this pathway is not immediately activated once DNA damage occurs. Hence, the HR is also known as a slow(er) repair pathway. The downregulation of BRCA2 lasts a few hours. Since blocking of HR by hyperthermia, when hyperthermia was applied before ionizing radiation, did not result in different cell survival compared to the opposite order of treatment, it may indicate that mild hyperthermia with temperatures ≤42 °C has minimal effects on the fast DNA repair pathway, the nonhomologous end-joining pathway (NHEJ). The NHEJ is considered an error-prone pathway, which is active during all phases of the cell cycle, and as a consequence this repair process can immediately respond when DNA double strand breaks occur [20]. The long duration of the radiosensitization found in our results could support this conclusion. Therefore, our results may suggest that, in these in vitro models, mild hyperthermia mainly affects the slow DNA repair pathway, the HR. However, we have also reported previously that if HR is blocked, in HR knock-out cells, hyperthermia is also seen to inhibit the NHEJ [30]. We have not seen a significant effect on blockage of the NHEJ in the current models.

All the in vitro data converge to the conclusion that a higher temperature 42 °C vs. 39 or 41 °C of HT and the shortest time interval (0 h) between ionizing radiation and hyperthermia results in the lowest cancer cell survival. In six different cervical cancer cell lines, all experiments were conducted independently and performed at least three times. Our results confirm the data reported in a study of Van Leeuwen et al. [27], in which a pronounced positive effect on tumor control was reported of shortening the time interval between radiotherapy and hyperthermia, without inducing more normal tissue toxicity. Moreover, Van Leeuwen et al. [31,32] also found a statistically significant positive impact of reducing the median time interval between radiotherapy and hyperthermia from 90 to 60 min on tumor control and overall survival in 58 locally advanced cervical cancer patients treated with radiotherapy and hyperthermia. HPV-positive cell lines are sensitive to hyperthermia because hyperthermia disrupts the interaction between HPV-protein E6 and p53, which results in cell cycle arrest and apoptosis [33].

Decades ago, similar experiments were performed on mammalian cell lines, in which Chinese hamster ovary (HA-1) and mouse mammary sarcoma (EMT-6) cells were investigated regarding the interaction between hyperthermia and ionizing radiation. Li and Kal concluded that in HA-1 cells the radiosensitizing effect of hyperthermia was larger when applied before ionizing radiation [34], but in EMT-6 cells the opposite sequence had a larger effect. We did not find a preference for a sequence to achieve the lowest cell survival. In the studies with HA-1 and EMT-6, cells were treated for 1 h at 43 °C with a high dose (6 Gy) of ionizing radiation, which can result in a significant amount of direct cell kill—for the ionizing radiation dose we used a range of 0 up to 8 Gy and to best simulate clinical practice, our highest hyperthermia dose did not exceed 1 h at 42 °C. Another explanation for the differences in sensitivity to ionizing radiation and hyperthermia between the previous study and our present study, may result from the biological difference in rodent sarcoma-like tumor cell lines HA-1 and EMT-6 and the human cervical carcinoma cell lines that we used. In our study we checked the cell survival after 14 days of treatment, which is basically clinically more relevant than an earlier time point.

In the present study we investigated the effectiveness of the time interval between ionizing radiation and hyperthermia in cervical cancer tumor cell lines and we found a that a short time interval resulted in more cell death. The sequence of applying both treatments did not make a significant difference. These data on sensitivity of cancer cell lines are a clear indication that the time interval between the treatments is an important factor. However, to define how radiotherapy and hyperthermia should be applied in the optimal way, it is a necessity to find the right balance between maximal effects on tumor tissue and minimal effects on the normal tissue. Therefore, to optimize the current clinical treatment regimes, we have to consider the effects on the normal tissue. The best way to study these effects would be in an *in vivo* model, in which the tumor microenvironment is taken along as well as the relevant healthy tissues around the target region. In patients, a short time interval has been shown to result in lower in-field recurrence and a longer overall survival [32]. This result was supported by the fact that most DNA double strand breaks had been already repaired in patient biopsies taken at two hours after a single dose of ionizing radiation. As a consequence, inhibiting DNA repair by hyperthermia is not effective after a longer time interval when all DNA breaks have already been repaired. Moreover, to define the optimal time interval between ionizing radiation and hyperthermia, a large and thorough investigation is needed. Therefore, an animal study would be a logical next step in finding the optimal balance between damaging the tumor and sparing the normal tissue. In addition animal models permit the study of the physiological effects of hyperthermia [35] and the effects of hyperthermia in the tumor microenvironment, which also contribute to radiosensitization [36]. In *in vivo* animal models, when there is a functional immune system, this permits the study of the effect of hyperthermia on the immune system [37], as both the tumor microenvironment and immunological effects are extremely important to take into account when studying the effectiveness of radiotherapy and hyperthermia.

In an *in vivo* model for mammary cancer heated to 42.5 °C, Overgaard (1980) found a slightly more pronounced enhancement in radiosensitization for hyperthermia treatment given before than after radiotherapy. He also found a more significant enhancement in radiosensitization for simultaneous treatment (time interval 0 h) compared to 0.5 h, which could also indicate that a faster DNA repair pathway was involved at that temperature [22]. These results are in line with our findings. However, the results from our current study are only applicable to cervical cancer cell lines. In order to find the effectiveness on other tumor models, additional cell lines of those specific tumor types need to be tested. Moreover, also for other tissue types, both in vitro and in vivo data are required before any conclusions on the optimal treatment of applying radiotherapy and hyperthermia can be made. Crezee et al. [38] explained that in vivo experiments are needed to assess the contributions to hyperthermic radiosensitization by reoxygenation and by direct elimination of the hypoxic tumor fraction. Dewhirst et al. [39] described that at relatively low tumor temperatures (39–40 °C), the effect on treatment outcome is dominated by reoxygenation of the tumor, rather than inhibition of DNA repair, which requires temperatures exceeding 41 °C. Hyperthermia applied before radiotherapy is expected to yield higher thermal enhancement ratios when reoxygenation is dominant, but also, a shorter time interval between hyperthermia and radiotherapy is expected to be more effective, as demonstrated in an *in vivo* tumor model by Overgaard (1980). Based on both clinical data [32] and preclinical data [22], the time interval between ionizing radiation and hyperthermia can influence treatment outcome. The effects when radiotherapy and chemotherapy are combined with hyperthermia could result in more tumor cell eradication, however, effects on normal tissue could be enhanced as well. Thus, in order to find the most optimal way of applying radiotherapy and hyperthermia, the effects of time interval between the two modalities should be taken into account, and the effects on both tumor and normal tissues are required.

## 4. Materials and Methods 

### 4.1. Cell Lines and Cell Culture

The cervical carcinoma cell lines, HPV16^+^ SiHa and Caski, HPV18^+^ HeLa and C4I, and HPV-negative C33A and HT3, were obtained from the American Type Culture Collection (ATCC). SiHa, HeLa, and C33A cells, were grown in EMEM (BioWhittaker/Lonza). Caski cells were grown in RPMI-1640 (Gibco-Brl Life technologies), C4I cells were grown in Waymouth’s medium (Gibco-Brl Life technologies), and HT3 cells were grown in McCoy’s 5a (Gibco-Brl Life technologies). All mediums contained 25 mmol/L Hepes (Gibco-BRL life technologies, Breda, The Netherlands) supplemented with 10% heat-inactivated fetal bovine serum (FBS) and 2 mmol/L glutamine. Cells were maintained in a 37 °C incubator with humidified air supplemented with 5% CO_2_. The cell division time of these cells was approximately 24 to 60 h.

### 4.2. Cell Treatments

Cells were treated with ionizing radiation and hyperthermia at 0, 1, 2, 3, or 4 h intervals between the two modalities, for either of the two sequences. For a time interval of 0 h, the second treatment started immediately after the first treatment was completed. Varying doses of ionizing radiation and temperatures of hyperthermia were used, depending on the assay. Hyperthermia was performed in a thermostatically controlled water bath (Lauda aqualine AL12, Beun de Ronde, Abcoude, The Netherlands) by partially submerging the culture dishes for approximately 1 h at 39, 41, or 42 °C. In order to check the temperature, thermocouples were placed in parallel culture dishes and the desired temperature (± 0.1 °C) was reached in approximately 5 min. Hyperthermia on all cells was performed in a 5% CO_2_/ 95% air atmosphere, both with an air inflow of 2 L/min. Ionizing radiation was performed with gamma-ray from a ^137^Cs source at a dose of about 0.5 Gy/min, and for survival curves, cells were irradiated with single dose up to 8 Gy, For cell cycle distribution and apoptosis level, cells were irradiated with 4 Gy, and for γ-H2AX staining, cells were treated with a single dose of 2 Gy. The doses were optimized for these specific assay in previous studies [40]. 

### 4.3. Cell Survival Assay 

Clonogenic cell survival on all six cervical cancer cell lines was studied to investigate the effect of different sequences and different time intervals when applying ionizing radiation in combination with hyperthermia. Cells were plated 4 hours prior to treatment with different doses (0, 2, 4, 6, and 8 Gy) of ionizing radiation and several time intervals (0, 1, 2, 3, and 4 h) between these two therapies were investigated. Furthermore, these set ups were tested on varying temperatures of HT (37, 40, 41, and 42 °C). Dishes were placed in an incubator with 5% CO_2_ at 37 °C until sufficiently large colonies were formed. Afterwards the medium was removed, before cells were washed with PBS. A mixture of 6% glutaraldehyde and 0.5% crystal violet was added for at least 30 min at room temperature. Subsequently, plates were washed with tap water and dried in normal air at room temperature. Colonies were counted under a light microscope. Surviving fractions were calculated by dividing the plating efficiency of treated cells by the plating efficiency of control cells with standard deviation [39]. Surviving fractions after dose D (S(D)/S(0)) were corrected for the cells treated with hyperthermia alone, at the desired temperature, and survival curves were analyzed to calculate values of the linear and quadratic parameters α and β, using SPSS [V25.0.01] (Chicago, IL, USA) statistical software by means of a fit of the data by weighted linear regression, according to the linear-quadratic formula: Ln(S(D)/S(0)) = (αD + βD²) [41]. Statistical analyses were performed using a two-way ANOVA with the software of GraphPad Prism version 8 (GraphPad Software, Inc. San Diego, USA).

### 4.4. Detection of DNA DSBs via γ-H2AX Staining

To study DNA damage in these cervical cancer cells, γ-H2AX staining was performed. Cells were seeded, at a density of 300,000 cells per coverslip, 24 h before treatment with ionizing radiation (2 Gy) and HT (37, 39, 41, and 42 °C) on sterile coverslips placed in culture dishes. Then, 24 h after treatment, cells were fixed for 15 min with PBS containing 2% paraformaldehyde for immunocytochemistry staining. After three times washing with PBS, cells were permeabilized during a 30 min incubation with TNBS (PBS containing 0.1% Triton X-100 and 1% FCS). Then, cells were stained with a primary mouse monoclonal antibody anti-γ-H2AX (Millipore, Merck, KGaA, Darmstadt, Germany, dilution 1:100 in TNBS for 90 min at room temperature). Cells were washed once with PBS and two times with TNBS before staining with the secondary goat antimouse-Cy3 (Jackson-Immunoresearch, dilution 1:100 in TNBS for 30 min at room temperature, in the dark). After washing cells two time with PBS and two times with TNBS, cells were incubated at room temperature with 50 µL Edu mix (Thermo Fisher Scientific, Massachusetts, USA). After a 60 min incubation under parafilm, cells were washed two times with PBS. Finally, vectashield-containing DAPI (Life technologies, Rockford, IL, USA) was dropped on the slide before placing the coverslips upside down on the slide. DSBs were scored under the fluorescence microscope. Statistical analyses were performed using a two-way ANOVA with the software of GraphPad Prism version 8 (GraphPad Software, Inc. San Diego, USA).

### 4.5. Cell Cycle Analysis

Experiments on cell cycle distribution were performed using the thymidine analogue 5-Bromo-2’-deoxy-uridine (BrdU, Sigma Aldrich, Merck, KGaA, Darmstadt, Germany). At 16 h after treatment, cells (42 °C hyperthermia, 4 Gy ionizing radiation) were incubated for 1 h with BrdU at 37 °C before washing with PBS, trypsinizing cells, and transferring them to 15 mL tubes. After centrifugation of cells at 1200 rpm for 10 min, cells were fixated in 2 mL PBS and 6 ml of 100% ethanol. The next day, cells were washed with PBS and centrifugated and the pellet was resuspended in pepsin-HCl (0.4 mg/mL, 0.1 N HCl) for 30 min at room temperature. Then, cells were washed using PBT (0.5% Tween-20, Sigma Aldrich USA in 0.5 l PBS) and after centrifugation, the pellet was resuspended in HCl (2 N, Merck, KGaA, Darmstadt, Germany) and cells were incubated 30 min at 37 °C. After washing once with PBT_b_, the pellet was incubated with the primary antibody rat anti-BrdU (Abcam, Cat. No ab6326, Cambridge, UK) diluted 1:100 in PBT_b_ (1% bovine serum albumin, Sigma, in PBT) for 60 min at 37 °C. After washing two times with PBT and once with PTB_g_, cells were incubated for 60 min at 37 °C with a secondary antibody IgG goat antirat FITC (Abcam, Cambridge, UK) diluted 1:100 in PBTg (1% normal goat serum, Dako, Glostrup, Denmark, in PBT). Eventually, propidium iodide (Sigma, Aldrich, USA) was added and cell suspensions were vortexed before analyzing samples with the flow cytometry (FACS Canto, BD Biosciences, San Jose, CA, USA). 

### 4.6. Apoptosis Assay

To study apoptosis after all treatment options, the Nicoletti assay [41] was performed. Directly after treatment, cells were cultured in a 37 °C incubator with the desired percentage of CO_2_. After 48 h, cells were collected and pellets were resuspended in Nicoletti buffer (0.1% w/v Sodium citrate, 0.1% v/v Triton X-100 in demi water, pH 7.4). Analyses were performed using flow cytometry (FACS Canto, BD Biosciences, San Jose, CA, USA). Statistical analyses were performed using a two-way ANOVA with the software of GraphPad Prism version 8 (GraphPad Software, Inc. San Diego, USA).

### 4.7. Statistical Analysis

Statistical analysis were performed with SPSS [V25.0.01] (Chicago, IL, USA) data are expressed as the mean with standard deviation of at least three independent experiments. Statistical analyses for cell survival, γ-H2AX staining, and apoptosis levels were performed using two-way ANOVA analysis using GraphPad Prism version 8 (GraphPad Software, Inc. San Diego, USA).

## 5. Conclusions

In summary, hyperthermia is an effective tumor sensitizer for ionizing radiation. However, optimizing therapies is important to find the best clinical outcome. Our in vitro results demonstrated that the time interval between hyperthermia and ionizing radiation is an important factor that determines the effectiveness of this combinational therapy. Moreover, our results may indicate that HPV-negative cell lines are already quite sensitive to ionizing radiation alone, while the HPV-positive cell lines require hyperthermia in order to reduce cell survival. The HPV16+ cell line seems to be more sensitive to the combination of hyperthermia and ionizing radiation, already at lower doses of ionizing radiation, compared to the HPV18+ cells. However, the survival fraction, cell cycle distribution, apoptotic levels, and DNA damage repair is more dependent on the time interval between the two treatments, rather than on the type of HPV. In conclusion: a shorter time interval between ionizing radiation and hyperthermia results in lower survival fractions than after a longer time interval, and the sequence between the two modalities did not demonstrate any significant differences on cervical cancer cell lines. However, studying the effects on normal tissue should be taken into account before the most optimal way of applying hyperthermia and radiotherapy can be determined.

## Figures and Tables

**Figure 1 cancers-12-00582-f001:**
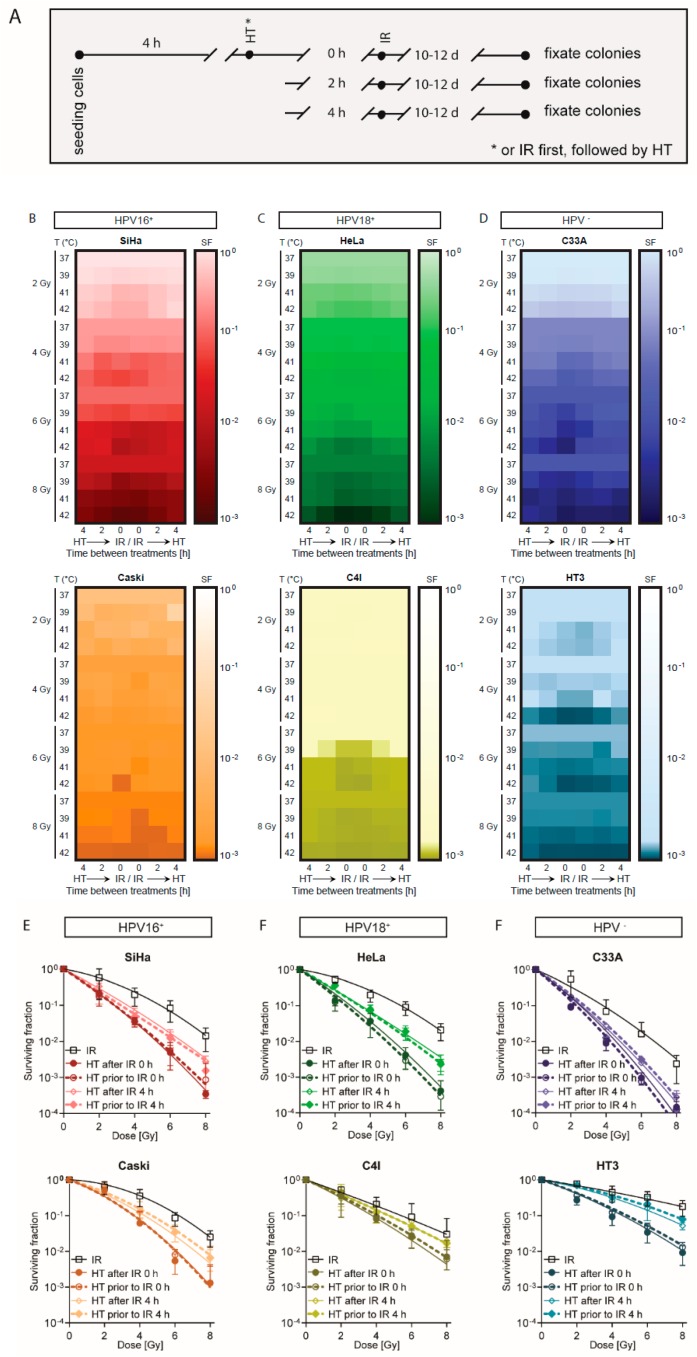
A short time interval between ionizing radiation and hyperthermia decreases cell survival. To study the effect of time interval (0, 2, or 4 h) between ionizing radiation (IR) and hyperthermia (HT), clonogenic assays were performed for six cervical carcinoma cell lines. (**A**) Schematic overview of applied treatments. (**B**–**D**) Survival fraction is demonstrated in heat maps: (B) HPV16^+^ cell lines SiHa and Caski, (C) HPV18^+^ cell lines HeLa and C4I, and (D) HPV-negative cell lines C33A and HT3—a darker color indicates lower cell survival. Per heat map, from top to bottom, different doses (2, 4, 6, and 8 Gy) of ionizing radiation are presented. Within one dose of ionizing radiation, cells treated with different temperatures of hyperthermia (37, 39, 41, and 42 °C). From left to right, within one heat map the left side shows results for hyperthermia applied before ionizing radiation with 0, 2, and 4 hour time interval, while on the right side ionizing radiation was applied before hyperthermia. Means of at least four experiments are presented. (**E**–**G**) Survival fraction after ionizing radiation (2, 4, 6, and 8 Gy) alone, and hyperthermia (42 °C) combined with ionizing with a 0 hour and 4 hour time interval between the two modalities, in both sequences. (E) HPV16^+^ cell lines SiHa and Caski, (F) HPV18^+^ cell lines HeLa and C4I, and (G) HPV-negative cell lines C33A and HT3. Means with standard deviation of at least four independent experiments are presented.

**Figure 2 cancers-12-00582-f002:**
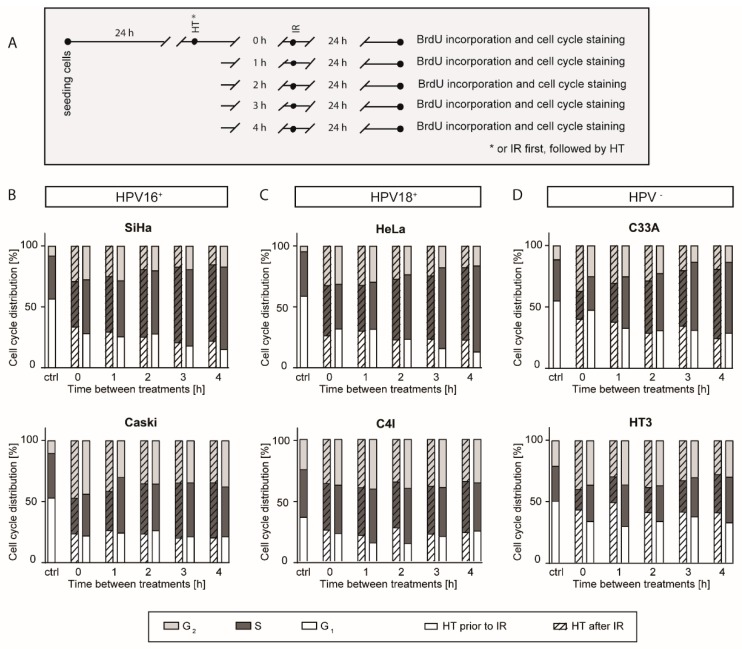
More pronounced G2 arrest after a short time interval between ionizing radiation and hyperthermia. Cell cycle distribution, using BrdU incorporation after different time intervals and sequences of ionizing radiation (IR) and hyperthermia (HT), was performed on six cervical cancer cell lines. (**A**) Schematic overview of treatment. (**B–D**) Cell cycle distribution of the cell lines, from left to right: HPV 16^+^ cell lines SiHa and Caski, HPV18^+^ cell lines HeLa and C4I, and HPV-negative cell lines C33A and HT3. Untreated samples are marked as control (ctrl). In the samples treated with a short time interval (0 h) between the two therapies, a more pronounced increase in G2 phase was observed and in some cell lines after a time interval (4 h) between the two therapies, the S phase was increased. However, cell cycle distribution was more dependent on the cell line than on the type of HPV. Means of at least four replicates are presented.

**Figure 3 cancers-12-00582-f003:**
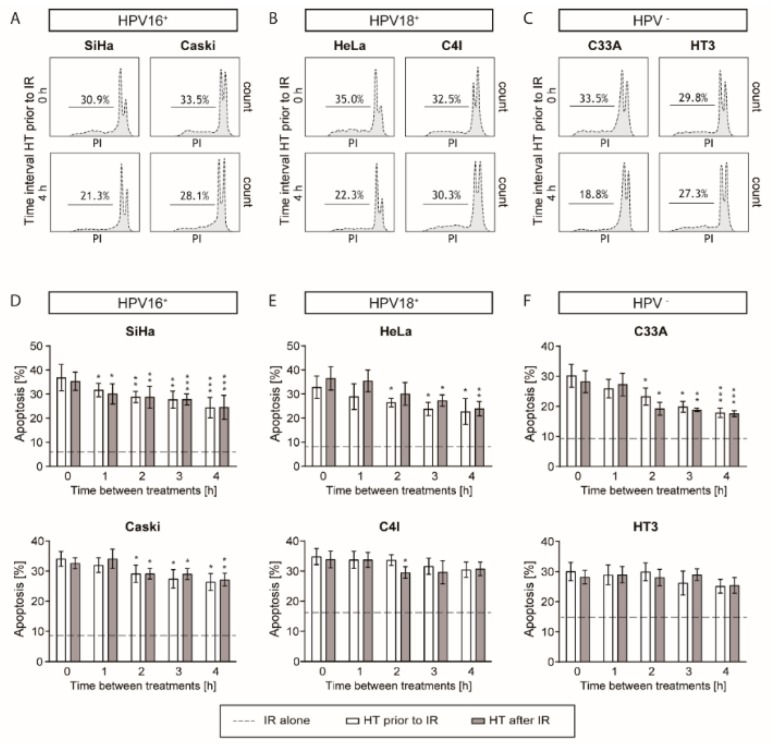
Higher levels of apoptosis were detected after shorter time intervals between ionizing radiation (IR) and hyperthermia (HT). Apoptosis levels were measured using the Nicoletti assay. (**A–C**) Representative flow charts of apoptosis levels of HPV16^+^, HPV18^+^, and HPV-negative cell lines. (**D–F**) Higher levels of apoptosis were observed after a short time interval between ionizing radiation and hyperthermia compared to a longer time interval. Combined hyperthermia with ionizing radiation had a higher level of apoptosis compared to ionizing radiation only, the dotted line shows the levels of cell apoptosis treated with ionizing radiation only. Applying hyperthermia before or after ionizing radiation did not result in any significant differences in SiHa, Caski, HeLa, C4I, C33A, and HT3 cell lines. Means with standard deviation of at least three replicates are presented. * *p* < 0.05, ** *p* < 0.01, and *** *p* < 0.001 indicate the difference between 0 and 1, 2, 3, and 4 h time intervals between ionizing radiation and hyperthermia, or the opposite order of treatments.

**Figure 4 cancers-12-00582-f004:**
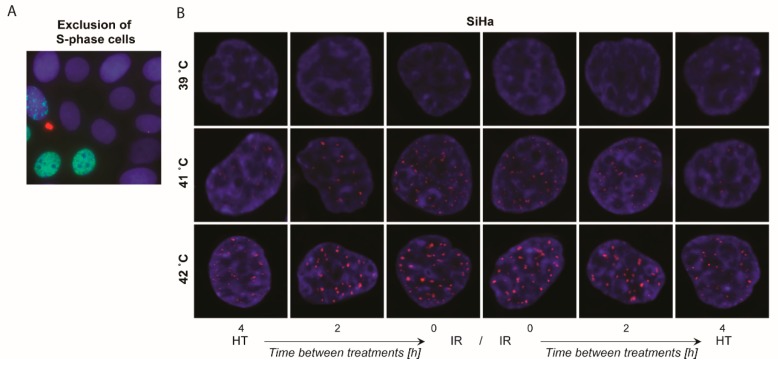
Higher levels of DNA damage were observed after treatment with a shorter time interval between ionizing radiation and hyperthermia. (**A**) Cells in S-phase were excluded to only count the γ-H2AX positive double strand breaks (DSBs) that were radiation-induced. (**B**) SiHa cells demonstrating the DNA damage using γ-H2AX foci—cells were fixed 24 h after treatments. (**C–E**) γ-H2AX foci of HPV16^+^, HPV18^+^, and HPV-negative cell lines between ionizing radiation (IR; 2 Gy) and hyperthermia (HT) after treatments with different time intervals, different sequences, and various temperatures. Means with standard deviation of at least three replicates are presented, with a minimum of 100 cells per replicate. * *p* < 0.05, ** *p* < 0.01, and *** *p* < 0.001 indicate the difference between 0 and 1, 2, 3, and 4 h time interval between ionizing radiation and hyperthermia, or the opposite order of treatments.

**Figure 5 cancers-12-00582-f005:**
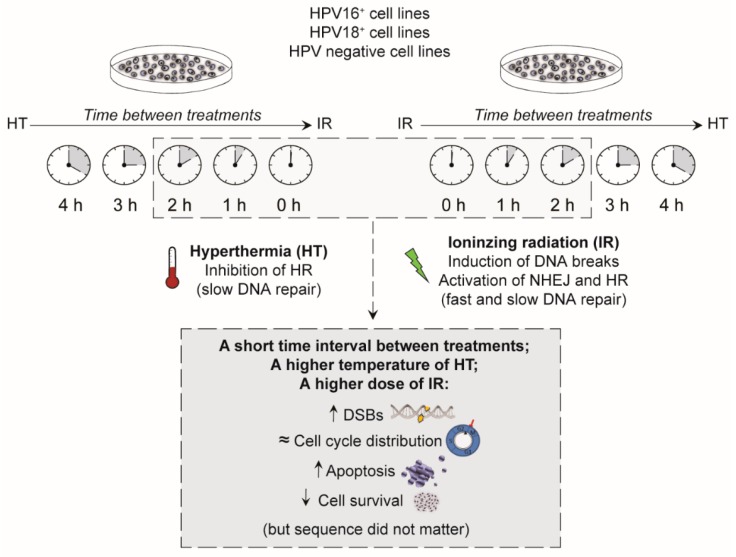
Summary. Effects of temperature, sequence, and time interval. A shorter time interval between ionizing radiation (IR) and hyperthermia (HT), a higher temperature of hyperthermia, and a higher dose of ionizing radiation were found to be of great importance in cell survival, by causing more residual DNA double strand breaks (DSBs) and higher levels of apoptosis. Either hyperthermia prior to ionizing radiation or hyperthermia after ionizing radiation did not matter on the number of DNA breaks, on the cell cycle distribution, levels of apoptosis, or on cell survival.

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
