# Peer review of "Radiosensitization by Hyperthermia: The Effects of Temperature, Sequence, and Time Interval in Cervical Cell Lines"

_cancers, 2020, doi:10.3390/cancers12030582_

Round 1

Reviewer 1 Report

Xionge and colleagues report the use of cervical cancer cell lines to evaluate hyperthermia as a mode of radiosensitization in vitro. While hyperthermia is not a standard clinical therapy for cervical cancer today, the promising data from clinical trials justify detailed studies to optimize the protocol for clinical treatment. As such, studies such as this are of high importance in establishing effective and uniform treatment protocols. The experimental setup for this study is ambitious and well presented. The manuscript is clearly written.

Major comments

  1. In order to establish a useful clinical protocol for cancer treatment, it is as important to show the effect of the treatment protocol on ‘normal tissue’. I appreciate that doing animal studies as the authors elude to themselves is a whole study on its own. But I do think that the authors should include one or two ‘cell lines’ more representative of normal tissue for comparison. One could for example use primary fibroblasts from a couple of different individuals. It is perhaps not necessary to carry out all the modes of investigation on these cells and it would suffice with survival assays and the most clinically relevant conditions (e.g. different temperature and time points with 2 Gy)
  2. Given the number of variables simultaneously tested in the survival assay, the heat map representation of data is a great choice. The data do convey the message of the manuscript regarding the effect of the time between the two treatments. On the other hand, the data suggest that the heating alone has a survival consequence which is also suggested by some previous studies. It is possible that when the two treatment are carried out too long apart, there is an additive tumor-killing contributed by the two modes of treatment, while with simultaneous treatment there is a synergistic effect where heating in fact functions as a radiosensitizer. To be able to conclude whether this is the case, a representation of the cell survival at different temperatures and different times and sequences shown here at a fixed radiation dose relative to the untreated sample (a single sample with no radiation and kept at 37 C) is required. Since the heatmap is missing 0 Gy and in the supplementary data the 0 Gy time-point is set to 1 for each temperature, this comparison is not possible to make. I suggest adding this for the dose of 2 Gy for all the cell lines in the supplementary section.   

Minor comments

  1. Statistical analysis of the cell cycle distribution presented in figure 2 is not shown.
  2. Why have the authors chosen 4 Gy for cell cycle and apoptosis analysis? The dose most commonly used in non-palliative treatment is approximately 2 Gy per fraction. The type of DNA damage, repair and mode of cell death is different for higher doses of radiation effecting both analyses.
  3. In their discussion, the authors attempt to present a possible mechanism for the differences seen in the different cell killing for the different time delays between the treatments. It is clear from the cell cycle data, and residual g-h2AX, that the mode of cell damage and DNA repair is different for the different time-gaps but I recommend shortening of the speculations on specifics of the mechanism since no experimental data are carried out to support this. I would also recommend against proposing an optimal temperature for hyperthermia, unless normal tissue experiments are presented. For example a dose of 8 Gy clearly is effective in tumor killing but cannot be use clinically. It may be that 42 °C together with radiation would result in too much normal tissue side effects.

Author Response

Point-to-point reply to the editorial recommendations and reviewers comments.

Mei et al. “Radiosensitization by hyperthermia: the effects of temperature, sequence and time interval in cervical cell lines”,
manuscript no. 726255

We would like to thank the expert reviewers for critically evaluating our manuscript and providing us with valuable criticism that has helped us to improve the quality of our work.

We have been able to address all points that have been raised in this review process.

Please find attached an overview of our response to the issues raised as well as how we have addressed them in our revised manuscript.

Reviewer 2 Report

The manuscript presents an interesting set of experimental data on the combined effects of hypothermia and irradiation on cancer cells. In my opinion, there are two issues which should be sorted out.

1.       Describing the combined effects of hypothermia and irradiation on cell survival, the authors summarised the results of their study on two graphs (Fig 1 and Suppl. Fig. 1). It is clear that high-dose exposure to ionising radiation significantly decreases the survival. However, from these graphs it does not follow that hypothermia statistically significant decreases the survival of irradiated cells. The alleged trend is visible on the heatmap, but there is no evidence that it is significant.

2.       The data on the combined effects of hypothermia and irradiation on apoptosis are summarised on Fig 2. According to the legend to this figure, it should present means with standard deviation. I could not find them in the submitted manuscript.

Author Response

(The authors gave the same response as above.)

Reviewer 3 Report

General comments:
Authors present an in vitro study on the combination of hyperthermia (HT) and radiotherapy (RT), using various HPV positive and negative human cervix cancer cell lines. Main goal of the study is to explore the importance of treatment sequence and treatment intervals on the exerted cytotoxic effects. Authors conclude that shorter intervals between therapies results in lower cell survival and increased DNA damage, whereas the sequence of treatments has no major relevance.
The study relates to a clinically relevant topic that deserve some attention. The design of the study is appropriate and the results have been correctly evaluated, however, there is a number of issues that should be considered by the authors before the study is ready for publication:

1. In all the figures of the manuscript, results on the combination treatments are shown. However, it is very relevant to show also outcomes of the single treatment modalities (at different doses and temperatures respectively) in order to compare the added value of combining the two treatments.
2. In the experiments on cell cycle distribution and apoptosis (Figures 2 and 3) a single radiation dose (4Gy) was chosen, whereas for the nuclear foci staining assay (Figure 4), they choose 2Gy. There is inconsistency in the chosen parameters between assays. Authors should argue on the choice of radiation dose for the assays.
3. Figure 3 is missing in the manuscript.
4. In Methods, radiation protocols are insufficiently described.
5. In this study authors check for the levels of apoptosis, however they do not check for induction of cell senescence. This is a particularly relevant things since senescent cells are not dividing but metabolically active and could contribute to tumor recurrence by SASP.
6. Ideally, this study should have been completed with an in vivo experiment showing therapeutic outcomes (tumor shrinkage) of combined treatments (at different time intervals) vs single treatments.

Author Response

(The authors gave the same response as above.)

Reviewer 4 Report

This is an exhaustive treatise on how various treatment variables in the combination of hyperthermia and radiotherapy, affect clonogenic cell survival and other radiobiologically relevant variables. Several different cervix cancer cell lines were used, some of which were HPV positive.  The latter is relevant for cervix cancer, particularly  in light of the author’s prior work showing that hyperthermia HPV positive cell lines are sensitive to hyperthermia, because hyperthermia disrupts the interaction between HPV-protein E6 and p53, leading to apoptosis. 

To a large extent, the results are not surprising.  Higher temperatures, longer heating times and higher radiation dose all result in lower survival. The closer in time that heating is done, relative to RT, the lower the survival.  These are all things that have been extensively validated pre-clinically before.  What makes this work different and worth reporting, though, is the additional detailed analysis of cell cycle effects, apoptosis, and double strand DNA breaks (gamma H2AX).  The combination of all of these variables enables the authors to tease out what the underlying mechanisms might be for the different effects seen. Additionally, the use of several different human cancer lines is important, because one can consider class effects (HPV status), as opposed to effects that might be dominated by a single cell line.

There is one major deficiency in the work, however. The statistical methods are not provided, as far as I can tell and even if they were, I do not think they were appropriate. It appears to me that the stats were done using T-tests, which is not appropriate for multivariate experiments.  The authors should consult with a biostatistician to figure out how to analyze the results, either by multivariate analysis or ANOVA.  As far as I can tell, there is not a biostatistician as a co-author on the paper.  At least if there is one, that person is not identified.

I also note that Figure 3 appears to be missing from this draft submission.

Is it important to note that this work was done entirely on cervix cancer lines and therefore may not be the same as other tumor lines?

I realize that standard of care for locally advanced cervix is the combination of hyperthermia and RT for your institution. But this is not the case in other parts of the world, where chemoRT is standard of care.  There have been some small studies wherein chemoRT was combined with hyperthermia for cervix cancer.  It might be important to at least speculate on how the results of this study might be affected by adding chemotherapy (typically cisplatin).

Specific Comments

HPV is introduced in the abstract, but the definition of HPV is not provided. Line 67 – pre-heating tumor increases perfusion and reduces hypoxia … Line 68 - The induction of free radicals after hyperthermia does not occur in the nucleus, to my knowledge. So there would not be any direct interaction with DNA damage or repair thereof. Line 80 – Increased hyperthermic cell killing is more related to acidic and nutrient deprived cells, rather than hypoxic cells. See early papers by Leeper's group on this. Also, Gerweck showed that hypoxic cells are not more sensitive to hyperthermia Line 83 – the English is awkward here. You can say "here we investigate..” or, "In this study..." Line 139 – You indicate that the stats methods were previously described, but they are not described as far as I can tell.

Author Response

(The authors gave the same response as above.)

Round 2

Reviewer 4 Report

This version of the paper is greatly improved. However, I find a few discrepancies that need to be corrected.  These are indicated below.

Line 61.  Oxygenation likely increases as a result of increased perfusion and a lowering of oxygen consumption rate.  Streffer showed many years ago that hyperthermia increases glycolysis (see Int J Hyperthermia, DOI - 10.3109/02656739709012386.  Second, hyperthermia induces HIF-1, which will push surviving cells toward glycolysis – see Moon et al., PNAS, 2010- DOI 10.1073/pnas.1006646107.  A change in vascular permeability has nothing to do with improving oxygen delivery. See Dewhirst and Secomb, Nature Reviews 2017- DOI 10.1038/nrc.2017.93.  For small molecules, transport is dominated by the concentration gradient across the vessel wall.

Line 79.  Here you indicate that pre-treatment with hyperthermia causes hypoxia by vascular expansion.  This makes no sense and in fact is not true, within the temperature range that is achievable with hyperthermia.  As I indicated above, hyperthermia causes a switch to glycolysis and this switch will reduce oxygen consumption rate.  Vascular damage does not occur within clinically achievable temperatures.  Even a small reduction in oxygen consumption rate will lead to a dramatic improvement in oxygenation. See Secomb et al, Acta Oncologica, 1995- DOI 10.3109/02841869509093981. Also, see Song et al, DOI: 10.1667/0033-7587(2001)155[0515:IOTOBM]2.0.CO;2

Line 345.  Because you are studying DNA damage repair, you surmise that all effects are related to this. But this may not be the only explanation.  There may be physiologic effects that are also important.  There could also be immunologic effects. There is the old saying that “if you have a hammer, everything is a nail!”. 

In your cover letter, your comments indicate that you are apparently unaware that there are clinical reports combining hyperthermia, radiation and cisplatin.  In fact, there are such papers, including a randomized phase III trial from Japan.  See DOI: 10.1002/cncr.21128, DOI: 10.1002/cncr.11475, DOI: 10.1080/02656736.2016.1213430

Author Response

Point-to-point reply to the editorial recommendations and reviewers comments.

Mei et al. “Radiosensitization by hyperthermia: the effects of temperature, sequence and time interval in cervical cell lines”,
manuscript no. 726255

We would like to thank the expert reviewer for critically evaluating our revised manuscript and providing us with valuable criticism that has helped us to further improve the quality of our work.

We have been able to address all points that have been raised in this review process.

Please find below an overview of our response to the issues raised as well as how we have addressed them in our revised manuscript:

Reviewer 4-2

This version of the paper is greatly improved. However, I find a few discrepancies that need to be corrected. These are indicated blow.

4.2.1  Line 61. Oxygenation likely increases as a result of increased perfusion and a lowering of oxygen consumption rate. Streffer showed many years ago that hyperthermia increases glycolysis (see Int J Hyperthermia, DOI - 10.3109/02656739709012386. Second, hyperthermia induces HIF-1, which will push surviving cells toward glycolysis – see Moon et al., PNAS, 2010- DOI 10.1073/pnas.1006646107. A change in vascular permeability has nothing to do with improving oxygen delivery. See Dewhirst and Secomb, Nature Reviews 2017- DOI 10.1038/nrc.2017.93. For small molecules, transport is dominated by the concentration gradient across the vessel wall.

Thank you for your valuable suggestions. These suggested papers show that hyperthermia enhances tumour perfusion and decreases oxygen consumption.

We have added these important findings in our revised manuscript with the appropriate references: text line 61 onward: “Hyperthermia can induce hypoxia-inducible factor-1 (HIF-1), a potent regulator of tumour vascularization and metabolism, which can push surviving cells toward glycolysis [9]. Hyperthermia can indeed increase glycolysis and subsequently reduce oxygen consumption rate [10-12]. Thus hyperthermia has been found to enhance tumour perfusion and decrease oxygen consumption and as a result, hyperthermia reduces tumor hypoxia [9].”

4.2.2  Line 79. Here you indicate that pre-treatment with hyperthermia causes hypoxia by vascular expansion. This makes no sense and in fact is not true, within the temperature range that is achievable with hyperthermia. As I indicated above, hyperthermia causes a switch to glycolysis and this switch will reduce oxygen consumption rate. Vascular damage does not occur within clinically achievable temperatures. Even a small reduction in oxygen consumption rate will lead to a dramatic improvement in oxygenation.

See Secomb et al, Acta Oncologica, 1995- DO 10.3109/02841869509093981.Also,

See Song et al, DOI: 10.1667/00337587(2001)155[0515:IOTOBM]2.0.CO;2

You are completely right. Mild hyperthermia can increase glycolysis, which will result in the reduction of oxygen consumption rate, and lead to tumour reoxygenation and thus contribute to radiosensitization by hyperthermia. We have changed this in our revised manuscript with the appropriate references: text line 82-83: “First, pre-heating the tumour induces reoxygenation by enhancing blood flow and by reducing oxygen consumption [15-17].“

4.2.3  Line 345. Because you are studying DNA damage repair, you surmise that all effects are related to this. But this may not be the only explanation. There may be physiologic effects that are also important. There could also be immunologic effects. There is the old saying that “if you have a hammer, everything is a nail!”. 

Our apologies, since we have only studied an in vitro setting with only cervical cancer cell lines, we did not address other factors such as physiological factors, contribution of HIF-1 or tumour reoxygenation. Obviously, in a more sophisticated model, and most definitely in in vivo animal models, or in patients, we have to take into account that physiological and immunological effects are of great importance on the effectiveness of radiotherapy and hyperthermia. We have discussed this in our revised manuscript and added appropriate references: text line 359 onward: “Therefore, an animal study would be a logical next step in finding the optimal balance between damaging the tumour and sparing the normal tissue. In addition animal models permit to study the physiological effects of hyperthermia [35] and the effects of hyperthermia in the tumour microenvironment, which are also contributing to radiosensitization [36]. In in vivo animal models, when there is a functional immune system, permit to study the effect of hyperthermia on the immune system [37], as both the tumour microenvironment and immunological effects are extremely important to take into account when studying the effectiveness of radiotherapy and hyperthermia.”

4.2.4  In your cover letter, your comments indicate that you are apparently unaware that there are clinical reports combining hyperthermia, radiation and cisplatin. In fact, there are such papers, including a randomized phase III trial from Japan. See  DOI:1002/cncr.21128, DOI:10.1002/cncr.11475, DOI: 10.1080/02656736.2016.1213430

Thank you for noticing this, there are indeed papers in which combination of full-dose radiotherapy, chemotherapy and hyperthermia showed at least a 15% improvement in overall survival. Thanks for your suggestions, these give us more ideas about new projects.
